# Genetic Characterization, Transmission Pattern and Health Risk Analysis of Intestinal Colonization ESBL-Producing *Escherichia coli* in Vegetable Farming Population

**DOI:** 10.3390/microorganisms12122646

**Published:** 2024-12-20

**Authors:** Fanghui Yao, Qian Zhao, Di Wang, Xuewen Li

**Affiliations:** Department of Environment and Health, School of Public Health, Cheeloo College of Medicine, Shandong University, Jinan 250012, China

**Keywords:** vegetable cultivation area, ESBL-producing *Escherichia coli*, intestinal colonization, CTX-M, transmission pattern

## Abstract

The surging prevalence rates of ESBL-producing *Escherichia coli* (ESBL-Ec) pose a serious threat to public health. To date, most research on drug-resistant bacteria and genes has focused on livestock and poultry breeding areas, hospital clinical areas, natural water environments, and wastewater treatment plants. However, few studies have been conducted on drug-resistant bacteria in vegetable cultivation. In this study, a total of vegetable farmers (n = 59) from six villages were surveyed. Fecal samples were collected from vegetable farmers; we also collected environmental samples, including river water, well water, soil, river sediment, vegetable surface swabs, and fish intestinal tracts. The ESBL-Ec intestinal colonization rate in vegetable farmers was 76.27%. PFGE results indicated two patterns of ESBL-Ec transmission within the vegetable cultivation area: among vegetable farmers, and among river water, river sediments, and vegetable farmers. Based on the phylogenetic analysis, three transmission patterns of ESBL-Ec outside the vegetable cultivation area were inferred: human–human, human–animal–human, and human–animal–environment. Twelve of the isolates carried closely related or identical IncF plasmids carrying *bla_CTX-M_*. Whole genome sequencing (WGS) analysis showed that ST569-B2-O134:H31 and ST38-D-O50:H30 were associated with high disease risk. We assessed the health risks of the farming population and provided a reference basis for public health surveillance and environmental management by monitoring the prevalence and transmission of ESBL-Ec in vegetable areas.

## 1. Introduction

Since the development of antibiotics, they have been widely used in animal breeding, hospital clinical practice, and agricultural activities. This increases the prevalence of antibiotic-resistant bacteria (ARBs) and antibiotic-resistant genes (ARGs) [1]. Antibiotic resistance is a common medical problem worldwide. It has been listed by the World Health Organization (WHO) as a major public health threat to humanity in the 21st century. *Escherichia coli* is the most abundant bacterial species in the human gut. Under normal conditions, it is symbiotic with the body; however, when immunity is low, it will cause infection within the body. Today, *Escherichia coli* has developed into a multi-drug resistant bacterium (MDR) that is resistant to a wide range of antibiotics [2]. *Escherichia coli*, which produces ultra-broad-spectrum β-lactamase (ESBL), is particularly severe. ESBL is an enzyme that hydrolyzes a variety of β-lactam antibiotics (including third- and fourth-generation cephalosporins), and ESBL-producing strains are also resistant to other antibiotics (aminoglycosides, tetracyclines, etc.) [3]. The WHO released an updated version of the 2024 Catalogue of Priority Bacterial Pathogens, in which Enterobacteriaceae bacteria resistant to third-generation cephalosporins were the key priority. According to the 2022 China Bacterial Drug Resistance Monitoring Report, the national average resistance rate of *Escherichia coli* to third-generation cephalosporins was 48.6%, a decrease of 1.4 percentage points from 2021, but still relatively high. ESBL *Escherichia coli* (ESBL-Ec) is clinically susceptible to a variety of infections, especially urinary tract infections and bloodstream infections [4,5], and the infections are characterized by limited therapeutic options, a high disease burden, and a poor prognosis. Therefore, focusing on the prevalence of ESBL-producing *Escherichia coli* is important for promoting global health.

To date, environments for the study of ARBs and ARGs are mostly livestock and poultry breeding areas, hospital clinical areas, natural water environments, and wastewater treatment plants (WWTP) [6,7,8,9]. However, relatively few studies have been conducted in vegetable cultivation areas [10]. ESBL-Ec and other drug-resistant bacteria were found in vegetable foods obtained from farmers’ markets, supermarkets, school cafeterias, and other public places [11,12,13]. Some studies have shown that fertilization with animal manure and irrigation with river and well water can introduce ARBs into vegetable cultivation areas [14]. The present study site is a large-scale vegetable cultivation area in China, and its vegetable trade within and outside the country may increase the risk of transmission of ARBs and ARGs.

ARBs and ARGs can circulate in the human–animal–environment cycle. Contaminated environments and animals can transmit ARBs and ARGs to humans and undergo intestinal colonization. Humans, in turn, can reverse-transmit ARBs and ARGs into the environment and animals through diverse human activities [15]. Vegetable cultivation areas serve as relatively confined work environments, and prolonged exposure to ARBs and ARGs through interior activities may increase the risk of intestinal colonization and transmission of ARBs and ARGs. Some studies have shown that there is a correlation between the colonization of ESBL-producing bacteria in the intestinal tract and infections in the body and that high colonization rates usually cause infections [16]. Farmers are the key to connecting the internal and external environments of the vegetable cultivation areas. Theoretically, humans, as key nodes in this transmission cycle, can effectively reduce the large-scale spread of ARBs and ARGs if the transmission pathways associated with humans are cut off.

Therefore, this study was conducted to investigate the prevalence of ESBL-Ec in the intestine and surrounding environment of farmers in an intensive vegetable cultivation area in Shandong Province, China, using a culturable experimental method combined with molecular analysis. Focusing on the study population of farmers, possible health risks were analyzed at a superficial level by characterizing the isolates using resistance and virulence genes, serotypes and MLST typing. Analysis of ESBL-Ec transmission patterns within and outside vegetable areas based on phylogenetic analysis, aimed to provide a reliable basis for better prevention and control of the wide spread of ARBs and ARGs.

## 2. Materials and Methods

### 2.1. Study Site and Samples Collection

A total of vegetable farmers (n = 59) from six villages within the vegetable cultivation area were surveyed, and fecal samples were collected from them. Survey participants were selected based on the distribution of greenhouses, with one owner of a greenhouse selected per kilometer. Fresh fecal samples were collected using sterile nylon swabs and sterilized centrifuge tubes and placed in Eswabs tubes (Copan, Brescia, Italy) for storage. Fecal samples were mixed and shaken in 5 mL of sterilized brain/heart infusion broth.

To further explore the relationship of ESBL-Ec transmission between humans and the environment, we collected both environmental and other samples including river water (n = 20), well water (n = 24), river sediment (n = 20), fish gut (n = 9), greenhouse soil (n = 87), and surface swabs of vegetables and melons (n = 106). River water and sediment samples were collected at 20 sampling points along a main stream flowing through the plantation. Each sample of river and well water was 800 mL and kept in sterile brown glass bottles. River sediment samples of >30 g each were collected with a stainless-steel spoon at the corresponding location of the river water sampling point. Soil and vegetable samples were collected from 87 vegetable greenhouses. Various samples were collected and processed as described previously [10].

Figure 1 shows the geographic location of the vegetable cultivation area and the distribution of sampling sites.

### 2.2. Screening of ESBL-Producing Escherichia coli

All samples were placed in sterilized brain heart infusion broth at 37 °C in a constant-temperature horizontal shaker for bacterial enrichment prior to screening for resistant bacteria. According to the Clinical and Laboratory Standard Study Guidelines (CLSI), we screened the ESBL strains in the samples by uniformly coating 100 μL of the bacterial solution on MacConkey’s selective medium spiked with 2 mg/mL of cefotaxime in a bacterial thermostat incubator for 24–48 h. Samples were then screened for ESBL strains. Eight colonies per plate were selected based on the color, morphology, and size of the colonies on the medium, or all were selected if there were fewer than eight colonies. After screening, matrix-assisted laser desorption ionization time-of-flight mass spectrometry (MALDI-TOF-MS) was used to identify the isolated ESBL strains. PCR experiments (primers are shown in the Table 1) were performed to determine the presence of β-lactam resistance genes in ESBL-producing *Escherichia coli* isolates, and the amplicons were subjected to one-generation sequencing to identify the CTX-M gene isoforms.

### 2.3. Antimicrobial Susceptibility Testing

The following 12 antibiotic drugs including meropenem (MEM), imipenem (IMP), ceftazidime (CAZ), cefotaxime (CTX), polymyxin B (PB), ciprofloxacin (CIP), tigecycline (TCG), amikacin (AMK), tetracycline (TE), fosfomycin (FOS), furotoxin (U), and piperacillin/tazobactam (TZP) were selected for drug sensitivity testing of ESBL-producing *Escherichia coli* isolates, except for PB and TCG, which were performed using the micro broth dilution method, and the rest were performed using the agar dilution method. Resistance phenotypes were determined by observing the minimum inhibitory concentration (MIC) of the experimental strains of each antimicrobial drug according to the Clinical and Laboratory Standards Institute guideline 2023 (CLSI) and the European Committee on Antimicrobial Susceptibility (EUCAST). *Escherichia coli* ATCC 25922 was the control strain.

### 2.4. Plasmid Transferability-Related Series of Experiments

The transferability of the β-lactam resistance gene was verified by splicing experiments using the sodium azide-resistant strain J53 as a recipient strain. The recipient and donor strains were mixed in LB broth at a ratio of 1:2 and incubated in a bacterial thermostat incubator at 37 °C for 12 h. Transplants were screened by LB agar plates spiked with 2 mg/L of cefotaxime and 350 mg/L of sodium azide. At the same time, LB agar plates supplemented with sodium azide were used to ensure that the donor bacteria did not grow or mutate. Detailed steps for S1 nuclease pulsed-field gel electrophoresis (S1-PFGE) and Southern blotting were performed as previously described [10]. The probes used in this study were *bla_CTX-M_* probes. S1-PFGE helps in the separation and analysis of large DNA fragments, while Southern blotting provides a way to detect and analyze specific sequences within those fragments.

### 2.5. Pulsed Field Gel Electrophoresis

DNA preparation for pulsed field gel electrophoresis (PFGE) was performed as described previously [17]. After digestion of agarose-embedded DNA with *Xba-I* at 37 °C for 2 h, electrophoresis was performed using a CHEFDR-III electrophoresis apparatus (Bio-Rad, Hercules, CA, USA) in 0.5× TBE buffer for 16 h, with the parameters set at 14 °C and pulse times of 2.16–63.8 s, and *Salmonella H9812* was used as the labeling strain. The electrophoresis results were used to map the PFGE fingerprints using BioNumerics software version 8.0, we used 85% as a threshold value, and isolates were considered related if the PFGE fingerprint similarity was >85% [18]. The transmission pathway of ESBL-Ec in this vegetable cultivation area was postulated by identifying the affinities of isolates from different media sources, and traceability analysis within the vegetable cultivation area was carried out for ESBL intestinal colonization by occupational personnel.

### 2.6. Whole Genome Sequencing and Biosignature Analysis

Since this study focused on farmers in vegetable cultivation area, we selected 45 fecal ESBL-Ec isolates for whole genome sequencing to analyze their genetic characteristics. DNA was extracted from ESBL-Ec by using the Gentra Puregene Yeast Extraction Kit and subsequently sequenced by Novogene (Beijing, China) using the Illumina NovaSeq 6000 platform (Illumina, San Diego, CA, USA). Sequencing results were assembled using SPAdes 3.11. Multilocus sequence typing (MLST), resistance genes, virulence genes, and serotypes. The datasets generated during the current study were deposited in GenBank (ID: PRJNA1178885).

### 2.7. Phylogenetic Analysis of Intestinal Colonization of ESBL-Ec

To further explore the potential sources of intestinal colonization of ESBL-Ec among farmers within the vegetable cultivation area, we screened and downloaded ESBL-Ec strains from the NCBI database for different regions, time (October 2012–March 2024), and sources in Shandong Province. At the same time, 21 ESBL-Ec isolates from the Jinan wastewater treatment plant that were sequenced by our group were included [7], and all of the above strains were phylogenetically analyzed using cgSNP. We performed a comprehensive phylogenetic traceability analysis of intestinal colonization by ESBL-Ec at different ranges.

## 3. Results

### 3.1. Detection of ESBL-Ec

A total of 429 ESBL isolates were obtained from 325 samples, and 57 ESBL-Ec strains were isolated (Table 2). Of these, 45 isolates were isolated from the fecal samples of farmers, with the highest detection rate of 76.27%. Six isolates were isolated from river water samples, two isolates were obtained from well water samples, one isolate from river sediment sample, one isolate from fish gut sample, and two isolates from soil samples. ESBL-Ec was not isolated from the surface-swab of vegetables samples.

### 3.2. Characterization of ESBL-Ec Resistance in Intestinal Colonization of Vegetable Farmers

#### 3.2.1. Resistance Phenotypes and Resistance Genes

The results of antimicrobial sensitivity experiments (Appendix A) showed that all isolates were resistant to cefotaxime with MIC values of ≥16 mg/L, and more than 50% of isolates were resistant to TE (95.6%, 43/45), CIP (93.3%, 42/45), FOS (84.4%, 38/45), U (60.0%, 27/45) and CAZ (57.8%, 26/45) resistance. Only two isolates were resistant to TZP (4.4%, 2/45). Notably, one isolate was resistant to IMP. All the strains were susceptible to AMK, MEM, TCG, and PB.

A total of five different β-lactam resistance gene combination types were detected. The most common types were *bla_CTX-M_* (23, 51.11%) and *bla_CTX-M_* + *bla_TEM_* (13, 28.89%), and two isolates carried *bla_CTX-M_*, *bla_TEM_* and *bla_SHV_* (Appendix A). Six *bla_CTX-M_* genotypes were detected, the most dominant of which was *bla_CTX-M-65_*. Clustering analysis of resistance genes carried by phylogenetic group, ST type, and serotype (Figure 2) show that *bla_CTX-M-55_* is often coexisting with *bla_TEM_*, *fosA3*, *tet(A)*, *aph(3″)-Ib*, and *bla_CTX-M-65_* is often coexisting with *oqxA* and *oqxB*. There is a correlation between ST type and *bla_CTX-M_*. All ST393 carry *bla_CTX-M-27_*, all ST155 carry *bla_CTX-M-55_*, and the majority of ST38 carry *bla_CTX-M-15_*.

#### 3.2.2. *bla_CTX-M_* Gene Skeleton and Plasmid Characterization

Gene skeleton refers to the various mobile genetic elements (MGEs), other resistance genes (ARGs), virulence genes (VFs), etc., present around the resistance gene *bla_CTX-M_*. A total of 19 different gene skeletons were identified for the six *bla_CTX-M_* subtypes (Figure 3), with *bla_CTX-M-65_* having the greatest diversity of gene skeletons (n = 6) and *bla_CTX-M-55_* having only one type of gene skeleton, which is often coexisting with *bla_TEM_*. The insertion sequence *IS5* is frequently found upstream and downstream of *bla_CTX-M-65_*, *bla_CTX-M-14_*, and *bla_CTX-M-27_* genes, and the transposon *Tn3* and *IS1380* insertion sequences are often found around the *bla_CTX-M-15_* resistance gene.

S1 nuclease PFGE and Southern blot hybridization experiments on *bla_CTX-M_* labeled with DIG using incompatible specific probes showed that a total of 33 ESBL-Ec strains of *bla_CTX-M_* were located on the plasmid, with sizes ranging from 30 kb to 250 kb; the majority of *bla_CTX-M_* (n = 12) was located on IncFII and IncFI plasmids of approximately 70 to 80 kb size, with *bla_CTX-M_* from one isolate being located on two IncFI plasmids (~60 and ~120 kb in size, respectively) (Appendix A).

### 3.3. Correlations Among MLST Types, Phylogenetic Groups and Serotypes

Seven phylogenetic cluster types, 17 ST types, and 24 serotypes were identified. Cluster A (n = 17) and cluster D (n = 12) were the dominant phylogenetic clusters. The Sankey diagram shows good correspondence among MLST, phylogenetic groups and serotypes of phylogenetic groups A, B1, B2, and D (Figure 4). The predominant ST type in phylogenetic cluster D is ST38 (n = 7), with serotype O50:H30. ST38-D-O50: H30 carries the *ast*A, *sep*A, and *pic* virulence genes. ST38-D-O80:H19 and ST38-D-O86:H18 carry the *ast*A virulence gene. The strain B2 is ST569-O134:H31 in phylogenetic cluster B2, carrying the *ibeA*, *tsh*, and *iha* virulence genes. Almost all strains carry the virulence gene *fim*H, and B2-ST569-O134:H31 carry more virulence genes than the other strains (Appendix A).

### 3.4. Analysis of the Transmission of ESBL-Ec for Intestinal Colonization by Vegetable Farmers

#### 3.4.1. Transmission Inside the Vegetable Area

PFGE fingerprinting revealed a high degree of genetic similarity among isolates from different sources within the vegetable cultivation area (Figure 5). A total of 52 ESBL-Ec isolates have 26 unique PFGE patterns (A–Z), and 12 groups of homologous isolates (each group consisting of two or more isolates) are identified based on the PFGE phylogenetic tree to identify genetically related isolates. Four groups of these isolates were obtained from different sources (groups A, B, E, and H, respectively), and the remaining eight groups were separated from the fecal samples of farmers from six different villages (groups C, D, F, J, K, M, N and P, respectively). The two main patterns of ESBL-Ec transmission within this vegetable cultivation area, inferred from the PFGE fingerprints, were between farmers, as well as among farmers, the river environment, and river sediments (Figure 6).

#### 3.4.2. Transmission Outside the Vegetable Area

The 2016 animal source isolates, the 2016 clinical source isolate, and the 2019 vegetable farmers’ isolates were closely related to each other. The 2018 WWTP isolates were closely related to the 2019 vegetable farmers’ isolates. The 2015, 2016, 2017, and 2019 clinical and 2019 vegetable farmers’ isolates were clonal strains (SNP < 10). Based on this, we summarized three possible patterns of ESBL-Ec transmission in Shandong Province: between farmers and hospital populations (Figure 7A); among farmers, animals, and hospital populations (Figure 7C); and among farmers, animals, and the environment (WWTP) (Figure 7B). Meanwhile, the 2023 clinical source isolates and 2020 animal source isolates were closely related to the 2019 vegetable farmers’ isolates (SNP < 10) (Figure 7E).

## 4. Discussion

The colonization rate of intestinal ESBL-Ec in vegetable farmers was as high as 76.27%. This suggests the presence of widespread colonization of ARBs and ARGs among farmers within the vegetable area, and the farmers could develop into a source of widespread ARBs and ARGs transmission within the cultivation area. It has been shown that ESBL colonizers are repositories of persistent ESBL transmission and that they are at risk of recurrent infection by colonizing strains. Fecal ESBL carriage has been associated with ESBL *Escherichia coli* infections, which may cause a wide variety of clinical infectious diseases (urinary tract infections, meningitis, and bloodstream infections) after intestinal colonization in humans, with the clinical outcomes of these infections being generally poor [19]. This suggests that we need to focus on monitoring and preventing ARB and ARG intestinal colonization in farmers in vegetable cultivation areas, as well as appropriate de-colonization measures for those with the ARB colonization.

Forty-three farmers with intestinal colonization of ESBL-Ec carry the CTX-M gene, and the dominant subtypes are *bla_CTX-M-65_* (28.3%) and *bla_CTX-M-15_* (21.7%). This is in general agreement with the finding that *bla_CTX-M-15_* (*bla_CTX-M-1_* group) and *bla_CTX-M-1__4_* (*bla_CTX-M-__9_* group) are more prevalent in humans, as reported by Seiffert et al. [20,21]. *bla_CTX-M-65_* is a variant of *bla_CTX-M-1__4_* belonging to the group *bla_CTX-M-__9_* [22]. *bla_CTX-M-65_* was first reported in a strain of Citrobacter fumigatus isolated from China in 2007. For the subsequent 17 years, the enzyme is currently spreading widely in epidemic proportions in several countries, including China, Korea, and the United States [7,23,24]. Notably, two ESBL-Ec isolates harboring *bla_CTX-M_*, *bla_SHV_*, and *bla_TEM_* show significantly higher levels of cefotaxime resistance than other isolates (MIC = 256). Current studies have not elucidated the relationship between bacterial drug resistance genes and drug resistance phenotypes. However, this result suggests a possible correlation between the number of resistance genes and the resistance level.

Possible sources of ESBL-Ec colonization of farmers’ intestines, as inferred from PFGE fingerprinting, are regionally attributed most to farmer-to-farmer, followed by river water and river sediments. A study analyzing the sources of acquired ESBL-producing *Escherichia coli* carried by a community-based population in The Netherlands showed that the most important attribute was person-to-person transmission, which accounted for 60.1% of the total transmission; 6.9% was attributed to secondary transmission by high-risk groups, including travelers, medical personnel, and farmers; followed by environmental, food, and animal sources [25]. The results of the present study are similar in comparison. This study also demonstrated that although the direct contributions of food, animals and the environment to the population acquisition of ESBL-producing *Escherichia coli* is relatively small, they expand the reservoir and enhance ESBL-Ec human-to-human transmission. In addition, river water samples had the highest detection rate of 30.00% in environmental samples, and previous studies have detected ESBL in water bodies, such as river water in Tai’an City and water for vegetable irrigation in Shaanxi [26,27]. This suggests that the aquatic environment is an important medium for the storage and transmission of drug-resistant bacteria. It is known that this vegetable cultivation area uses river and well water for vegetable irrigation and manure as a fertilizer to fertilize fields, which also provides a realistic basis for our presumed two possible patterns of ESBL-Ec transmission.

Outside the vegetable cultivation area, the possible transmission patterns of ESBL-Ec in Shandong Province, inferred based on phylogenetic analysis, were human–human, human–animal–human, and human–animal–environment. WWTPs are the major sources of antibiotic resistance genes [28]. A study of a wastewater treatment plant in eastern China showed that the ESBL-Ec isolate from the wastewater treatment plant was more than 95% homologous to the ESBL-Ec isolate from the nearby river [7]. Further analysis revealed that the gut-colonized ESBL-Ec isolates are closely related to WWTP isolates (SNP < 10). This suggests clonal transmission of ESBL-Ec between the two sites. Combined with the geographic relationship between the two areas and the nearby river, the river runs west from the city where the WWTP is located, and the vegetable cultivation area is a necessary place for the river to pass through, which provides a solid basis for the above inferences. Animal feces are often used as fertilizers in this vegetable cultivation area, which may account for the clonal transmission of ESBL between humans and animals. In terms of isolation time, clonal transmission of ESBL-Ec has continued to occur in Shandong Province since 2020. In particular, the strain F19 from the present study showed significant clonal transmission with *Escherichia coli* collected in 2023 from a hospital in Shandong Province. (Retrieve from the project PRJNA1036031). Close contact with colonizers increases the risk of infection and will eventually introduce ARBs and ARGs into hospitals [29].

Groups B2 and D *Escherichia coli* are usually extraintestinal pathogenic *Escherichia coli* (ExPEC), whereas, Groups A and B1 are usually considered intestinal commensals [30]. Strains of Groups B2 and D show high levels of virulence (Appendix A). ST38-D-O50:H30 strains carry the heat-stable enterotoxin 1 (*ast*A) virulence gene encoding heat-stable toxin 1 (EAST1), virulence gene sepA associated with enterotoxigenic *Escherichia coli* (ETEC), and virulence gene *pic* associated with enteroaggregative *Escherichia coli* (EAEC). astA is frequently found in foodborne pathogenic *Escherichia coli*. Although ESBL-Ec was not detected in vegetable samples in this study, the risk of disease in raw vegetables should not be ignored. ESBL and Ampc were detected in vegetables, especially green leafy vegetables, in a study of fresh vegetables in Israeli markets [12]. In another study of produce from Edo State, Nigeria, ESBL was detected in raw vegetable salad (20%, 12/60) [31]. ST569-B2-O134:H31 carries invasive brain endothelial protein A (*ibe*A), the avian pathogenic *Escherichia coli*-associated virulence gene *tsh*, and the adhesion virulence factor *iha*. IbeA is important invasive protein, it can cross the blood–brain barrier and is highly correlated with strain pathogenicity [32]. In addition, ST569-B2-O134:H31 carries *bla_CTX-M-65_* with an insertion sequence *IS5* upstream and downstream. *IS5* is associated with the mobilization and transmission of resistance genes among different bacteria [33]. Almost all the strains harbored the adhesive virulence gene *fim*H. A study on urethral pathogenic *Escherichia coli* (UPEC) showed that the type 1 bacterial hair adhesin *fim*H binds to mannose on the bladder surface and mediates bladder colonization [34]. Therefore, we inferred that *fim*H facilitates ESBL-Ec colonization in the guts of farmers. This suggests that strains of Groups B1, B2 and D that colonize the gut have the potential to evolve into pathogenic *E. coli*, such as EAEC, ETEC, and neonatal meningitis *Escherichia coli* (NMEC).

Twelve of the forty-five ESBL-Ec isolates in this study carried IncF plasmids containing *bla_CTX-M_*, which could be successfully spliced into *E. coli J53* by a splice transfer assay. Five of these strains carried *bla_CTX-M-55_*, a genotype often carried by the IncF plasmid. This is consistent with the findings of ESBL-Ec isolates from fecal samples from a broiler farm in Brazil [35]. IncF plasmids are often isolated from human and animal *Enterobacteriaceae*, and current studies have shown that IncF plasmids carrying the *bla_CTX-M_* resistance gene have been found in various bacteria [36]. The IncF plasmid not only provides a vector for the *bla_CTX-M_* resistance gene but also enhances the propagation capacity and stability of the gene [37].

The study site is a typical intensive vegetable cultivation area in north-central China. Farmers in the area are exposed to double risks of infection and transmission. Vegetables are traded across the country and worldwide. Therefore, we assess the health risks of farmers by monitoring the ESBL-Ec transmission in vegetable cultivation areas. Eventually, the goal of effective and timely control of ARB and ARG prevalence will be realized, and the safety of the cultivating population and public health will be guaranteed.

## 5. Conclusions

Our results show the presence of extensive intestinal colonization of ESBL-Ec in vegetable farmers. The two modes of transmission of ESBL-Ec within vegetable areas are among vegetable farmers, as well as among rivers, river sediments, and vegetable farmers. The three modes of transmission outside vegetable area are human–human, human–animal–human, and human–animal–environment. This suggests that a variety of human–animal–environmental activities increase the risk of ESBL-Ec transmission. The IncF plasmid plays an important role in the spread of *bla_CTX-M_* within the vegetable area. ST569-B2-O134:H31 and ST38-D-O50:H30 carry virulence genes associated with high disease risk, posing a potential health threat to vegetable farmers. This suggests that it is necessary to monitor and take measures to reduce the intestinal colonization of ESBL-Ec and de-colonize those who have been colonized to reduce the health risk to the population and to control the long-distance persistent transmission of ESBL-Ec.

## Figures and Tables

**Figure 1 microorganisms-12-02646-f001:**
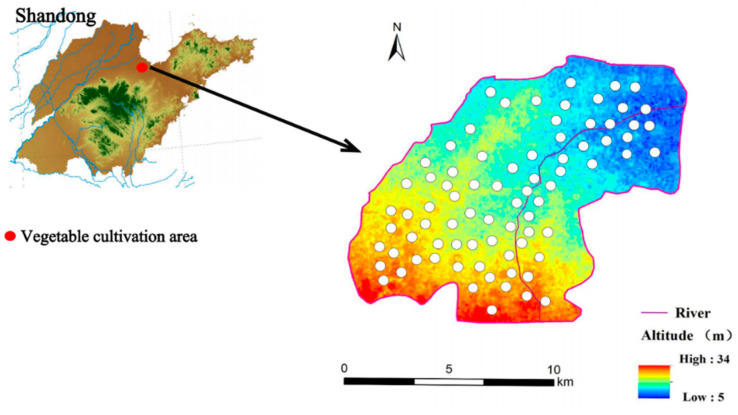
Vegetable cultivation area sampling map (white dots are greenhouse locations).

**Figure 2 microorganisms-12-02646-f002:**
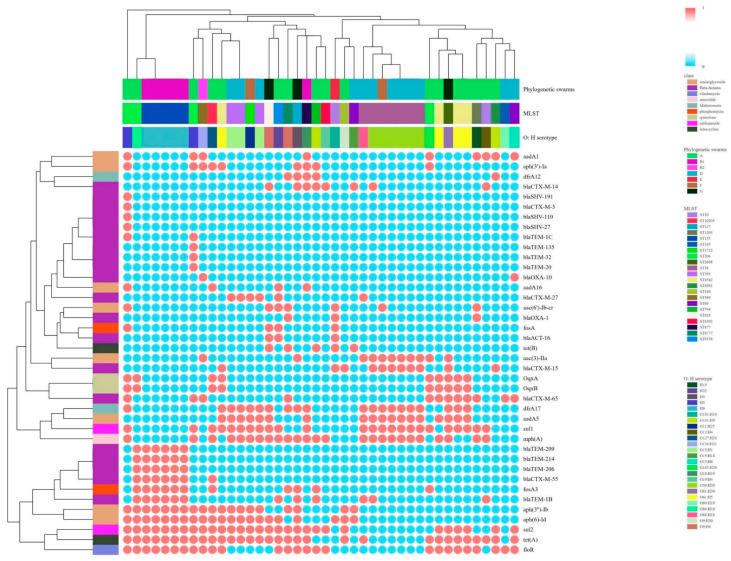
Clustering by phylogenetic group, MLST type and serotype to analyze the status of resistance genes carried by intestinal colonized ESBL-Ec isolates (red is carrying the resistance gene; blue is not carrying the resistance gene).

**Figure 3 microorganisms-12-02646-f003:**
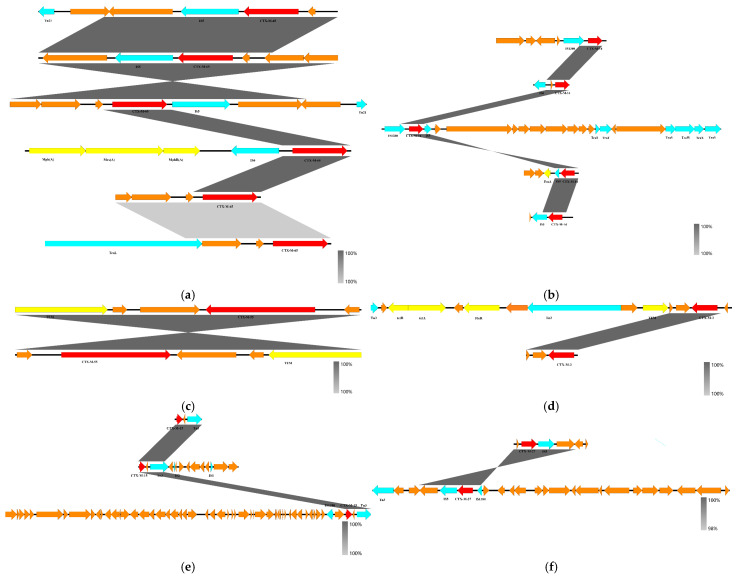
Different subtypes of CTX-M resistance gene skeletons (red arrows are CTX-M resistance genes, blue arrows are mobile genetic elements, yellow arrows are other resistance genes, and orange arrows are other genes): (**a**) gene skeleton of *bla_CTX-M-65_*; (**b**) gene skeleton of *bla_CTX-M-__14_*; (**c**) gene skeleton of *bla_CTX-M-__5__5_*; (**d**) gene skeleton of *bla_CTX-M-__3_*; (**e**) gene skeleton of *bla_CTX-M-__1__5_*; (**f**) gene skeleton of *bla_CTX-M-__27_*.

**Figure 4 microorganisms-12-02646-f004:**
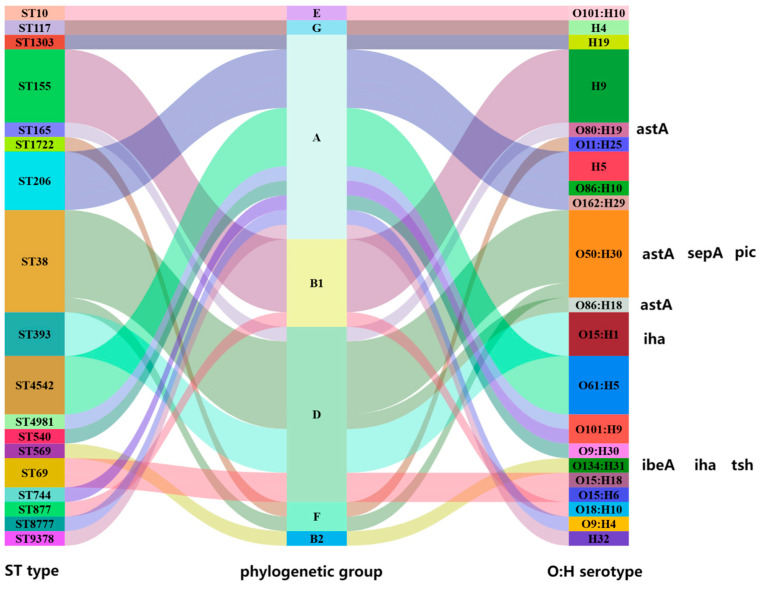
Phylogenetic clusters, MLST typing and serotype correlation analysis of intestinal colonized ESBL-Ec (right side is labeled as carrying the virulence gene).

**Figure 5 microorganisms-12-02646-f005:**
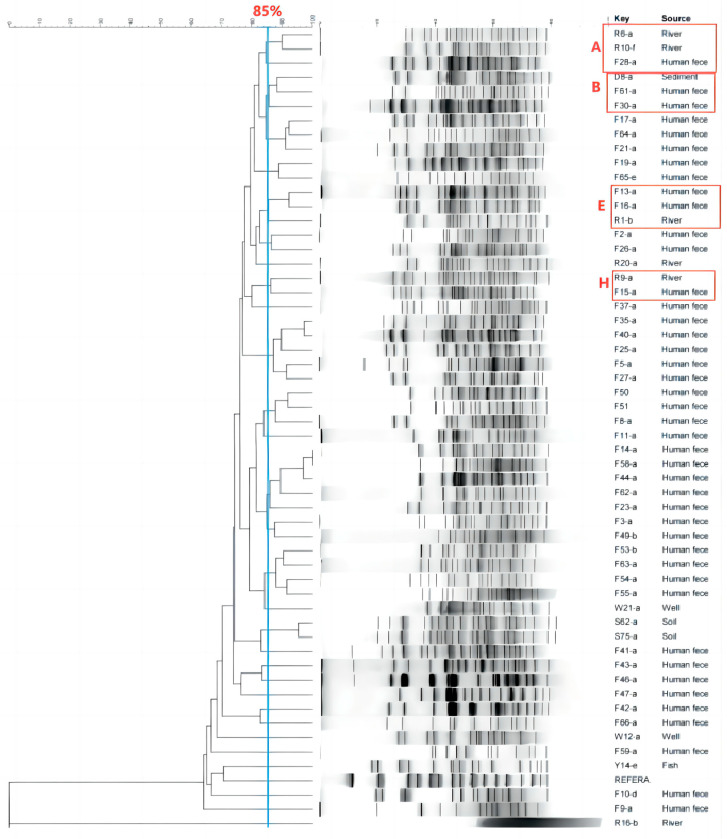
PFGE fingerprints of different samples of ESBL-Ec isolates within the vegetable cultivation area.

**Figure 6 microorganisms-12-02646-f006:**
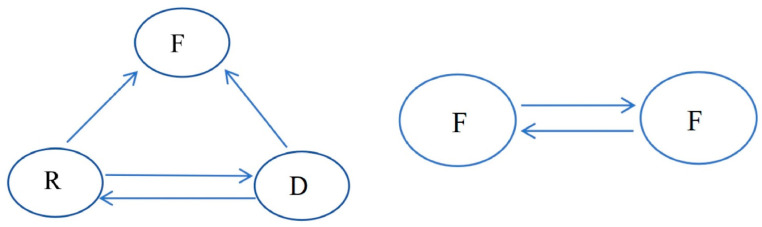
Transmission patterns of ESBL-Ec in vegetable cultivation environments identified based on PFGE analysis (F: cultivator feces; R: river water; D: river sediment).

**Figure 7 microorganisms-12-02646-f007:**
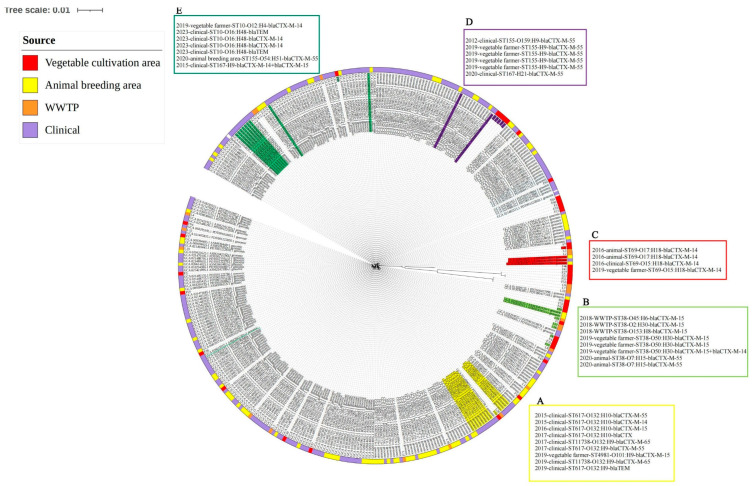
SNP phylogenetic analysis of ESBL-Ec isolates from different times and sources with farmer gut-colonized ESBL-Ec in Shandong Province, October 2012–March 2024 (labels outside the phylogenetic tree are labeled for strains (SNP < 10) for time, source, ST type, serotype, and resistance genes).

**Table 1 microorganisms-12-02646-t001:** Primers for β-lactam genes.

Primer Genotype	Primer Sequence	Clip Size	Annealing Temperature
CTX-M-F	TTTGCGATGTGCAGTACCAGTAA	544 bp	56 °C
CTX-M-R	CGATATCGTTGGTGGTGCCATA		
TEM-F	TTGATCGTTGGGAACCGGAG	861 bp	55 °C
TEM-R	TCCGCCTCCATCCAGTCTAT		
SHV-F	TGCTCATCATGGGAAAGCGT	861 bp	55 °C
SHV-R	ATCTCCCTGTTAGCCACCCT		
OXA-F	TCCTGTAAGTGCGGACACAA	762 bp	53 °C
OXA-R	TGGGAAAACTGGTGCAGGAT		

**Table 2 microorganisms-12-02646-t002:** Detection of ESBL-Ec from different samples in vegetable cultivation area.

Source	Sample Type	Number of Samples	ESBL-Ec	Detection Rate (%)
Human	Human feces (F)	59	45	76.27
Environment	River (R)	20	6	30.00
	Well (W)	24	2	8.33
	River sediments (D)	20	1	5.00
	Soil (S)	87	2	2.30
	Fish intestines (Y)	9	1	11.11
	Vegetable swab (V)	106	0	0.00
	total	325	57	17.54

## Data Availability

The original contributions presented in this study are included in this article/Appendix A.

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
