# Peer review of "Genetic Characterization, Transmission Pattern and Health Risk Analysis of Intestinal Colonization ESBL-Producing Escherichia coli in Vegetable Farming Population"

_microorganisms, 2024, doi:10.3390/microorganisms12122646_

Round 1
Reviewer 1 Report
Comments and Suggestions for Authors
I found the article well-written and interesting, with a clear research design and significant results. However, I have several suggestions for the authors to improve the manuscript. Below, I have outlined detailed feedback organized by section.
Title
- Avoid the use of abbreviations in the title to ensure clarity and accessibility for a broader audience.
- Explain all abbreviations, such as ESBL, ESBL-Ec, PFGE, and WGS, at their first mention.
- Include results regarding plasmids, as these are currently not mentioned but appear relevant to the study.
Introduction
- L38-40: Rewrite the sentence for clarity. An enzyme can confer resistance to a strain; however, the enzyme is not resistant. Consider rephrasing for accuracy.
Materials and Methods
- L84: Use "samples" instead of "sample" to match the plural context.
- L90: Remove "so" to improve readability.
- L1.6: Correct "MALTI" to the appropriate term ("MALDI").
- L110: Add a legend and title for the referenced table.
- L112-116: Add a main verb to complete the sentence.
- L118: Avoid repetition of the verb "to determine" for better flow.
- Throughout Paragraph 3.2, use the past tense instead of the present tense for consistency in reporting results.
- Paragraph 2.4: Provide additional details regarding S1-PFGE and Southern blotting. Specify the probe used for hybridization and clarify why the blotting was performed.
- Use italics for XbaI and Salmonella for proper formatting.
- L149: Ensure the sentence includes a verb to make it grammatically complete.
- In general, more methodological details should be provided for readers to improve reproducibility.
Results
- L168: Use "samples" instead of "sample."
- L183: Write gene names in lowercase.
- L189: Explain the expression "gene environment" for clarity.
- L205: Provide more information about the specific probes used.
- L217: Add the missing main verb to complete the sentence. Additionally, specify the analyzed virulence genes.
- Figure 4: Improve the image quality and add a more detailed legend to enhance understanding.
- L244-245: Include more details about the number and source of isolates. A table summarizing this information would be particularly helpful.
Discussion
- The discussion is clear and concise, but it does not address plasmids. Incorporating a mention of plasmid-related findings would provide a more comprehensive interpretation.
- Check the use of verb tenses for consistency.
- Ensure that gene names and similar scientific terms are italicized where appropriate.
Conclusion
- The manuscript lacks a conclusion. I suggest adding a brief conclusion that summarizes the key findings, their implications, and potential future directions.
Final Note
These revisions aim to enhance the clarity, completeness, and overall quality of the manuscript. I hope the authors find this feedback constructive and helpful for improving their work.
Author Response
请参阅附件

Reviewer 2 Report
Comments and Suggestions for Authors
This work has an element of originality since it focuses on vegetables and the antibiotic treatments that may eventually be present in these environments and the consequences in terms of mobility of resistance genes that they may have on susceptible human populations. The work is well planned and executed and the results shown are consistent to support the conclusions of the work.
There are some elements that must be considered in this study
The color scale in Figure 2 is illegible. It is recommended that you modify that part of the figure and include larger letters to be able to differentiate the details of the data shown in the figure.
The work indicates that the transfer in which river water is involved could be a relevant transmission route. In this sense, it would be worth considering that distribution in these ecosystem conditions would lead to a significant reduction in selection pressure as a consequence of the low concentration of the antibiotic, which could lead to a loss of the carrier plasmid. In this sense, it would be appropriate to analyze the presence of genetic factors that could be selected jointly in the same extrachromosomal elements (plasmids) and that would contribute to the maintenance of resistance genes in the bacteria object of this study.
I understand that it would be convenient to include some reference that demonstrates that the presence of siderophores or other elements associated with resistance genes in the same plasmid contributes to the retention of resistance despite the low concentration of antibiotics in the medium.
Considering the above, I understand that this work could be published with minor corrections
Reviewer 3 Report
Comments and Suggestions for Authors
This is original and unique research, and it was a pleasure to read such a work that is rich in results yet not too extensive because the authors skillfully provided what is most important.
Vegetable cultivation areas are scarcely screened for antibiotic-resistant bacteria (ARB) and antibiotic-resistant genes (ARGs), so this investigation is very important because the results could be used to prevent or decrease the risk of ARB and ARGs transmission among humans, animals, and their environment.
Minor comments:
-- The abbreviation WGS analysis is mentioned only in the Abstract. Please consider its explanation.
-- Abstract: Lines 17-18 “among vegetable farmers” is repeated twice.
-- The abbreviation ESBL (Extended Spectrum Beta Lactamase (ESBL) should be introduced at first mention (Introduction, Lines 37-38).
-- In M&M, subchapter 2.4. (Line 129) the abbreviation S1-PFGE is first mentioned, but not explained.
-- Lines 92-93: “Various samples were collected and processed as described previously[10]”. Authors are usually asked to write a short description in such situations anyway, so please do that.
-- Ref. 10 is incomplete; necessary data are missing; please add: 273:116370, doi: 10.1016/j.envpol.2020.116370).
-- Could you provide the images of the Southern blot hybridization results (white-black versions) in a better resolution?
Round 2
Reviewer 1 Report
Comments and Suggestions for Authors
The authors addressed all my requests improving the manuscript.
regards